# Uncertainty Assessment of Hyperspectral Image Classification: Deep Learning vs. Random Forest

**DOI:** 10.3390/e21010078

**Published:** 2019-01-16

**Authors:** Majid Shadman Roodposhti, Jagannath Aryal, Arko Lucieer, Brett A. Bryan

**Affiliations:** 1Discipline of Geography and Spatial Sciences, School of Technology, Environments and Design, University of Tasmania, Hobart 7018, Australia; 2Centre for Integrative Ecology, School of Life and Environmental Sciences, Deakin University, Burwood 3125, Australia

**Keywords:** uncertainty assessment, deep neural network, random forest, Shannon entropy

## Abstract

Uncertainty assessment techniques have been extensively applied as an estimate of accuracy to compensate for weaknesses with traditional approaches. Traditional approaches to mapping accuracy assessment have been based on a confusion matrix, and hence are not only dependent on the availability of test data but also incapable of capturing the spatial variation in classification error. Here, we apply and compare two uncertainty assessment techniques that do not rely on test data availability and enable the spatial characterisation of classification accuracy before the validation phase, promoting the assessment of error propagation within the classified imagery products. We compared the performance of emerging deep neural network (DNN) with the popular random forest (RF) technique. Uncertainty assessment was implemented by calculating the Shannon entropy of class probabilities predicted by DNN and RF for every pixel. The classification uncertainties of DNN and RF were quantified for two different hyperspectral image datasets—Salinas and Indian Pines. We then compared the uncertainty against the classification accuracy of the techniques represented by a modified root mean square error (RMSE). The results indicate that considering modified RMSE values for various sample sizes of both datasets, the derived entropy based on the DNN algorithm is a better estimate of classification accuracy and hence provides a superior uncertainty estimate at the pixel level.

## 1. Introduction

Assessing and mapping the state of the Earth’s surface is a key requirement for many global researches in the context of natural resources management [1], natural hazards modelling [2,3], urban planning [4,5] etc., where all these mapping products need to be validated [6,7]. With the initiation of more advanced digital satellite remote sensing techniques, accuracy assessment of emerging methods has received major interest [6]. The conventional way to report classification and/or prediction of map accuracy is through an error matrix estimated from a test dataset, which is independent of the training process [8]. Accuracy metrics such as Cohen’s Kappa coefficient [9], overall accuracy (OA) [7] and class-specific measures such as user’s and producer’s accuracies are usually estimated based on an error matrix [10]. However, it is not clear how these accuracy metrics relate to per-pixel accuracy [11] as these types of accuracy metrics are incapable of understanding the spatial variation of classification accuracies despite its importance in modelling spatial phenomena [12,13].

Different approaches have been proposed to characterise the quality of classified maps at the local scale [8]. One method is to apply empirical models to link classification accuracy (dependent variable) to different independent (predictor) variables, such as land cover class [14,15]. As the dependent variable is dichotomous (i.e., classified correctly or not), logistic regression is the most frequently applied algorithm for this purpose. Another approach to characterizing map quality at the local scale involves spatial interpolation of classification accuracy of the test dataset [16]. The most recent approach is introduced by Khatami et al. [8], built on Stehman [17]. Here, a per-pixel accuracy prediction is implemented by applying different accuracy prediction methods based on four factors, including predictive domain (spatial or spectral), interpolation function (constant, linear, Gaussian, and logistic), incorporation of class information (interpolating each class separately versus grouping them together), and sample size. The fourth and most popular approach [8] is to use the probabilities of class memberships or prediction strength (i.e., tree votes in the random forest (RF) or probabilities in neural networks) as indicators of classification uncertainty. The idea is that for a certain pixel, the greater the probability of class membership for a given labelled class, the lower the uncertainty associated with that class and analytical functions can be used to quantify the uncertainty measures instead of using only the membership value of the most probable class. Examples of these functions include ignorance uncertainty [18], Shannon entropy [19,20], and α-quadratic entropy and maximum probability [21], where entropy summarizes the information from membership values of all classes.

Uncertainty assessment techniques can provide an uncertainty map as a spatial approximator of classification accuracy, which can be used to locate and segregate unreliable pixel-level class allocations from reliable ones. In addition, this approach is independent of test data availability. This uncertainty assessment may be implemented using two types of classification approaches: unsupervised schemes using no training dataset [22,23], and supervised schemes [19,24,25,26]. Although unsupervised approaches can be applied regardless of the training dataset availability (i.e., by applying unsupervised algorithms), their relevant uncertainty assessment results may be misleading due to incorrect classification of pixels. In terms of supervised methods, various algorithms have been applied to evaluate the uncertainty of correct/incorrect classified pixels including RF as one of the most popular algorithms. RF [27,28] has a rich and successful history in machine learning including applications in hyperspectral image classification [29,30,31,32,33] and uncertainty assessment [34,35,36]. It has been demonstrated to outperform most state-of-the-art learners when it comes to handling high-dimensional data [37], such as hyperspectral image datasets. Nonetheless, we assumed that considering high-dimensional hyperspectral data, newly emerging deep learning algorithms may be efficient for uncertainty assessment, but they have been rarely applied for this purpose. On the other hand, the deep learning algorithms have also been found to be more accurate than traditional algorithms, especially for image classification [38,39,40]. Further, with multiple layers of processing, they may extract more abstract, invariant features of data, which is considered beneficial for uncertainty assessment studies.

Uncertainty assessment techniques have been repeatedly applied to assess the quality of hyperspectral image classification [23,41,42]. While deep learning has attracted broad attention as a classification algorithm [43,44,45,46], it has not been applied to uncertainty assessment of hyperspectral image classification nor compared to other methods. Thus, here we aim to apply deep neural network (DNN) for uncertainty assessment of correct/incorrect classification for every pixel and then compare it with RF. Due to its high performance in uncertainty assessment studies, the RF algorithm provides an appropriate benchmark for comparing the performance of uncertainty assessment derived from deep learning. This paper aims to explore, quantify and compare the capability of DNN and RF algorithms for uncertainty assessment of hyperspectral imagery using two different hyperspectral datasets. To this end, by applying DNN in this study, we compare the uncertainty assessment of hyperspectral image classification using probability values derived from deep learning neurons and popularity votes of RF trees combined with uncertainty values using Shannon entropy.

## 2. Methods and Dataset

### 2.1. Method

This study followed two major steps (Figure 1). In step 1, the whole dataset was randomly divided into training (50%) and test data (50%). For each dataset, the hyper-parameters of the optimum DNN and RF algorithms (Table 1) were configured using a 5-fold cross-validation of the training data in the pre-processing stage. This was done only for hyper-parameters with a significant effect on the datasets and the remaining hyper-parameters were kept at the default values. Although test data were always a constant sub-set of the whole dataset, the training procedure was done using different portions of training sample (i.e., 10%, 20%, …, 100%) to assess the effects of training sample size in uncertainty assessment. Thus, the training sample itself was sliced into 10 equal random portions, and then applied for training the tuned algorithms. The algorithms were then trained 10 times each, from 10% to 100%, every time by a 10% increase of training samples, i.e., x = {10%, 20%, …, 100%}, where *x* is a set of applied training samples. Here, the test dataset was always the same. In addition, to achieve more consistent results and to account for sensitivity analysis, each algorithm was applied in five consecutive runs, where the sampling strategy was the same but the locations of initial sampling seeds (i.e., random training (50%) and test data (50%)) were modified by a different random function. As the hyper-parameters of the DNN and RF algorithms were optimised using a validation sample, they were not modified for the other sample sizes. Here, for both DNN and RF, the probability of belonging to each possible class was estimated for every pixel and used to compute the uncertainty of classification for the pixel using Shannon entropy [20], where entropy represents uncertainty in this research [8].

In step 2, for a better demonstration of classification performance considering the low and high uncertainty values, we mapped the uncertainty outputs along with the mode of correct/incorrect classified test pixels for all applied training samples (i.e., from 10% to 100%). Whenever an optimised algorithm is applied in the context of uncertainty assessment, the uncertainty value for a correctly classified pixel should be minimised (i.e., “0”) while it should be maximised (i.e., “1”) for misclassified pixels. Thus, we then calculated root mean square error (RMSE) of every prediction implemented by each algorithm [20] to quantify the degree of deviation from this optimum state. For this purpose, entropy values were normalised between 0 and 1. This whole process was implemented in R [47] using three major packages namely “H2O” [48], “randomforest” [49], and “entropy” [50].

#### 2.1.1. Supervised Uncertainty Assessment Approach

The most popular and accurate way of uncertainty assessment is based on a supervised scheme using a machine learning algorithm. Here, we implemented a model that can assess the uncertainty values of a classified hyperspectral image containing various class labels. We first collected ground truth data labelled with their class categories such as corn, grass, hay, oats, and soybean. During training, the algorithm was provided with a training example and produced a response in the form of a vector of probabilities, one for each class. Then, the best-case scenarios would be the highest probability score for one class and the lowest possible probability score for the other remaining classes. The least desirable case, on the other hand, would be equal probability scores for all the existing class labels (Figure 2). We then computed the uncertainty of probability scores for all potential class labels for a pixel by using entropy. An ideal algorithm, for uncertainty assessment, is not only capable of classifying input data with the highest possible accuracy but also capable of producing class labels with low uncertainty for correctly classified pixels and vice versa.

In this study, the uncertainty derived from deep learning neurons and popularity votes of RF trees was quantified using Shannon entropy [51]. Entropy summarizes the information from membership values of all classes using Equation (1):(1)$$e_{x}=-\overset{h}{\underset{i=1}{\sum }}P_{i}\log_{2}P_{i}$$where P_i_ is the probability of class membership for h class labels. Further, the selection of the logarithm base is unimportant, as it only affects the units of entropy [25].

#### 2.1.2. Deep Neural Network (DNN)

The deep learning algorithm applied in this research is based on R studio deep neural network (DNN) from H2O package [48], which is a feed-forward artificial neural network, trained with stochastic gradient descent using backpropagation. Here, multiple layers of hidden units were applied between the inputs and the outputs of the model [52,53,54].

Each hidden unit, j, typically uses the logistic function β the closely related hyperbolic tangent is also often used and any function with a well-behaved derivative can be used) to map its outputsing y_j_ total input from x_j_:(2)$$y_{i}=\beta \left(x_{j}\right)=\frac{1}{1+e^{-x_{j}}}$$

For multiclass classification, such as our problem of hyperspectral image classification, output unit j converts its total input, x_j_, into a class probability, P_j_, by using a normalised exponential function named “softmax”:(3)$$P_{j}=\frac{\exp\left(X_{j}\right)}{\sum_{h}\exp\left(X_{h}\right)}$$where h is an index over all classes. DNNs are discriminatively trained by backpropagating derivatives of a cost function that measure the discrepancy between the target outputs and the actual outputs produced for each training case [55]. When using the softmax output function, the natural cost function C is the cross-entropy between the target probabilities d and the softmax outputs, P:(4)$$C=-\underset{i}{\sum }d_{j}\ln P_{j}$$where the target probabilities, typically taking values of one or zero, are the supervised information provided to train the DNN algorithm.

#### 2.1.3. Random Forests as a Benchmark

To measure and quantify DNN performance for uncertainty assessment of hyperspectral classification, we implemented the RF algorithm applied to the same datasets [49]. The RF algorithm provides an appropriate benchmark for assessing the performance of the DNN scheme because of its high performance found in hyperspectral data classification [30,31,32,56,57]. RF is also computationally efficient and suitable for training datasets with many variables and can solve multiclass classification problems [58]. We compared the uncertainty assessment results of DNN and RF using two different datasets.

#### 2.1.4. RMSE of Uncertainty Assessment

RMSE is the standard deviation of the residuals (prediction errors). Here, RMSE demonstrates standard deviation of prediction for correct and erroneous estimates of test dataset. In other words, it explains how concentrated the data are around the line of best fit considering entropy of correct and erroneous estimates:(5)$$\mathrm{RMSE}=\sqrt{\sum_{i=1}^{n}{\left(e-o\right)}^{2}/n}$$where *e* represents the estimated entropy value from “0” (minimum entropy value) to “1” (maximum entropy value) after normalisation; *o* represents classification result for the observed values, which is “1” for erroneous predictions and “0” for correct answers. Here, RMSE is applied as a goodness of fit for uncertainty assessment results. Therefore, the best-case scenarios would be those classification cases where the algorithm is at both the maximum confidence and accuracy (*e* = 0 and *o* = 0) or minimum confidence and minimum accuracy (*e* = 1 and *o* = 1). The worst-case scenarios, however, occurs when the algorithm is at minimum confidence and maximum accuracy (*e* = 1 and *o* = 0) or vice versa (*e* = 0 and *o* = 1). Table 2 demonstrates the intuitions behind the proposed RMSE.

### 2.2. Datasets

In this study, two widely used hyperspectral datasets including the Salinas [59,60,61] and Indian Pines [59,62,63] image datasets were used (Table 3) and divided into validation, train and test samples (Figure 3). Both datasets contain noisy bands due to dense water vapour, atmospheric effects, and sensor noise. These datasets are all available at http://www.ehu.eus/ccwintco/index.php?title%20=%20Hyperspectral_Remote_Sensing_Scenes.

The Salinas image consists of 224 bands and each band contains 512 × 217 pixels covering 16 classes comprising different sub-classes of vegetables (nine sub-classes), bare soils (three sub-classes) and vineyard (four sub-classes). It was recorded by AVIRIS sensor over the South of the city of Greenfield in the Salinas Valley, CA, USA on October 9, 1998. This dataset is characterised by a spatial resolution of 3.7 m, and the spectral information ranges from 0.4 to 2.5 µm. As shown in Figure 3, the ground truth is available for nearly two-thirds of the entire scene. We used 204 bands, after removing bands of the water absorption features.

The Indian Pines dataset is also an AVIRIS image collected over the Indian Pines test site location, Western Tippecanoe County, Indiana, USA on June 12, 1992. This dataset consists of 220 spectral bands in the same wavelength range as the Salinas dataset; however, four spectral bands are removed as they contain no data. This scene is a subset of a larger scene and it contains 145 × 145 pixels covering 16 ground truth classes (Figure 3). The ground-truthing campaign consists of approximately 10,000 samples which are distributed over the area of 2.5 km by 2.5 km. The ground truth data were collected by walking through the fields in the image. Plant species, as well as some more characteristics, were recorded along with photos of sites in the field. In the present research experiment, 20 spectral bands were removed because of the water absorption phenomena and noise.

## 3. Results

### 3.1. Salinas Simulation Experiments

The results of uncertainty assessment for the Salinas dataset using DNN and RF are presented in Figure 4. However, to avoid redundancy in the representation of the results, only half of the achieved uncertainty images are displayed (i.e., 10%, 30%, 50%, 70% and 90%). Regardless of the classification scheme and/or training sample size, classes 8 (8: Grapes_untrained) and 15 (15: Vinyard_1) belonged to the highest uncertainty level among all the available class labels. For both algorithms, this was followed by concentration of incorrect predictions within the high-uncertainty areas, which are identified as false values in the mode of correct/incorrect classified test data based on all training samples from 10% to 100%.

RF and DNN were comparable in terms of achieved OA of classification for the majority of sample sizes, while the areas covered with the high uncertainty values were less obvious within DNN results. This was observable for all corresponding sample sizes. Further, to quantify the capabilities of DNN and RF for uncertainty assessment and potential application as an estimate of accuracy, we then calculated the root mean square error (RMSE) of every sample size (Equation (5)) applied for image classification for each algorithm. Following the mapping of uncertainty values, which represent uncertainty levels, we plotted the RMSE of the classification (*y*-axis) of the test data for various training sample sizes (*x*-axis). Here, lower RMSE values indicate better estimates of uncertainty (Table 2), and vice versa. For the Salinas dataset, RMSE values for the DNN algorithm were lower than RF values for all sample sizes while RMSE values derived from RF are more consistent (Figure 5).

Further, to better understand the capability of uncertainty measures as an estimate of accuracy, we plotted the correspondence between mean class uncertainty (i.e., entropy) and class accuracy (Figure 6). Nonetheless, to avoid unnecessary repetition of results, only the 50% training sample was plotted, which confirmed the accuracy of classification within the majority of image classes will be reduced by an increase in the uncertainty of pixels belonging to these classes and vice versa. In accordance with Figure 4, it was also demonstrated classes 8 and 15 of Salinas dataset with the highest mean uncertainty values belong to the least accurate estimation.

### 3.2. Indian Pines Simulation Experiments

The results of uncertainty assessment for the Indian Pines dataset using DNN and RF were similar to those for the Salinas dataset. For both DNN and RF, classification uncertainty was reduced for larger training samples while the OA values of classification increased. However, these phenomena were less obvious for RF compared with those for DNN (Figure 7). In addition, the improvement of OA values with an increase in training sample size was more distinctive than that for the Salinas dataset. Remarkably, for every corresponding sample size, DNN was not only the more accurate algorithm but also displayed fewer pixels with high uncertainty values. The mode of correct/incorrect classified pixels demonstrated almost the same pattern for both algorithms while there were fewer misclassified pixels within the results of DNN algorithm.

The higher accuracy of DNN elevates the quality of implemented uncertainty assessment for locating correct/incorrect classifications for this dataset. Nonetheless, to quantify the difference in the quality of uncertainty the assessment between the two algorithms, the RMSE values were estimated for every training sample. The RMSE values also confirmed the superiority of DNN for the majority of training sample sizes in a way that less uncertainty was estimated for correct classified pixels while incorrect classified pixels were identified by more levels of uncertainty. However, the same as the Salinas dataset, RMSE values derived from five consecutive simulation runs of RF are more consistent. This can be easily observed by comparing the difference between the minimum and maximum RMSE values for each sample size that is observable in Figure 8. Obviously, DNN is coupled with more variation between minimum and maximum RMSE values for almost all different sample sizes.

Finally, the correspondence between mean class uncertainty (i.e., entropy) and class accuracy of Indian pine dataset is demonstrated in Figure 9 for 50% of training sample size using both DNN and RF algorithms. Similar to Salinas dataset results, the achieved results of Indian Pines demonstrated a negative relationship between uncertainty and accuracy for the majority of class labels.

## 4. Discussion

### 4.1. Comparing the Quality of Uncertainty Assessment Based on RMSE

With reference to the fact that both DNN and RF algorithms may achieve an OA above 70%, even for the minimum portion of training sample size (i.e., 10%), it was expected one algorithm may perform a better uncertainty assessment if it successfully limits the high-uncertainty areas to the spatial vicinity of incorrectly classified pixels while highlighting the remaining areas as low uncertainty. This is regardless of achieved OA, although the RMSE values derived from five consecutive runs of each algorithm indicate that results of uncertainty assessment using RF is more consistent compared with DNN. Nonetheless, comparing the results of uncertainty assessment, for both utilised datasets and every corresponding sample sizes, demonstrates that areas of high uncertainty values were less abundant within the results of DNN algorithm compared with that for RF algorithm (Figure 4, Figure 5 and Figure 7). This may be due to the fact that DNN is optimized to reduce the difference between the predicted distribution and the true data generating distribution by minimizing the cross-entropy of the two probability distributions [64,65]. Therefore, the uncertainty assessment derived from DNN algorithm was superior to RF combined with better OA for these two datasets. However, more studies using different datasets are still required for generalizing the results.

### 4.2. Quality of Uncertainty Assessment for Different Sample Sizes

For both algorithms and both datasets, larger training samples were found to be more beneficial for uncertainty assessment. The RMSE of uncertainty estimates, which was applied as a goodness of fit to assess the quality of uncertainty maps, decreased from the initial (10%) to final (100%) training sample sizes (Figure 5 and Figure 8). However, this improvement was more obvious for DNN compared with that for RF. This may be due to different formulations of RF and DNN algorithms, which are affecting the performance of the two algorithms for uncertainty assessment. Usually, the training sample size has a crucial role in classification accuracy [66]; thus, it will also affect the uncertainty assessment process. The increased training sample size will typically increase the performance of an algorithm from random sampling [67,68], but not all algorithms will be improved at the same level with a larger sample size. Although RF can also benefit from a larger training sample by extracting more binary rules [69], DNN may achieve a better performance. For DNN, the ratio of uncertainty assessment improvement followed by larger training sample size and more accurate classification depends on the abundance of contextual information per-pixel in the target dataset [70]. As many extensive experimental results confirm the excellent performance of the deep learning-based algorithms matched with rich spectral and contextual information [71], our study suggests this is also beneficial to increase the training sample to achieve a better uncertainty assessment result.

### 4.3. Uncertainty vs. Accuracy

The existing uncertainties at different stages of the classification procedure influence classification accuracy [66,72]. Therefore, understanding the relationships between the classification uncertainty and accuracy is the key successful contribution to an estimate of accuracy for image classification. Although a low uncertainty classification instance is accompanied with high accuracy, some exceptions may apply to the high uncertainty, which usually belongs to low accuracy estimates. Thus, incorrect predicted class labels are usually located inside high-uncertainty areas with very few exceptions within low-uncertainty regions while correct classified pixel overlay the low uncertainty areas (Figure 4 and Figure 7). In this research, for both applied datasets, the existing correspondence between uncertainty and accuracy was better identified using the DNN algorithm. Having said that, in our study, DNN is demonstrated more potential in uncertainty assessment for hyperspectral image classification. Following accurate classifications combined with minimising high-uncertainty areas, DNN not only offers a lower rate of RMSE but also offers a higher contrast between low and high uncertainty areas.

At a wider scale, considering mean class uncertainty against the class accuracy of test data, it was revealed that usually a lower uncertainty value of a class is followed by a higher accuracy (Figure 6 and Figure 9). In other words, as low uncertainty indicates the probabilities of potential class labels for a pixel are not equal (i.e., unimodal distribution). This simply specifies that based on the available distribution of potential labels and their probability values (Figure 2), defined by either deep learning neurons or tree votes, usually one of the potential class labels (i.e., 16 labels for each applied datasets) has a significant preference to be selected as the estimated label. Accordingly, the concentration of low uncertainty values corresponding to every pixel of the desired class label is anticipated by an acceptable accuracy of classification. In terms of higher values of mean uncertainty for a class, the class accuracy will be reduced due to the abundance of high uncertainty estimates within that class.

## 5. Conclusions

Due to the weaknesses of the traditional approaches of map accuracy assessment based on a confusion matrix, many uncertainty assessment approaches are being developed as accuracy estimates. In terms of supervised methods, we compared DNN with RF, where an estimate of accuracy is defined by the entropy of all potential probabilities/votes toward different class labels for a pixel, as an uncertainty measure. In this research, entropy was applied to encode the measure of uncertainty, which is applicable to any dataset including hyperspectral image datasets. Considering the results of uncertainty assessment, for both Salinas and Indian Pines datasets, DNN outperformed RF for the purpose of uncertainty assessment. However, the superiority of DNN algorithm was more obvious when applying the Indian Pines dataset, as well as larger training sample sizes. This was due to less-abundant high uncertainty values throughout the classified dataset compared with RF for every corresponding training sample size while having a comparable or better OA. Nonetheless, the achieved uncertainty maps of DNN can facilitate the application of hyperspectral image classification products by alerting map users about the spatial variation of classification uncertainty over the entire mapped region as an estimate of accuracy.

## Figures and Tables

**Figure 1 entropy-21-00078-f001:**
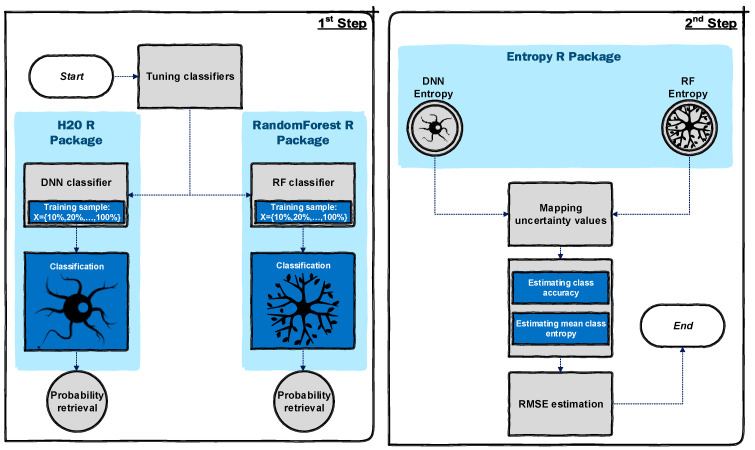
Flowchart of methodology implementation labelled with the main R packages utilized.

**Figure 2 entropy-21-00078-f002:**
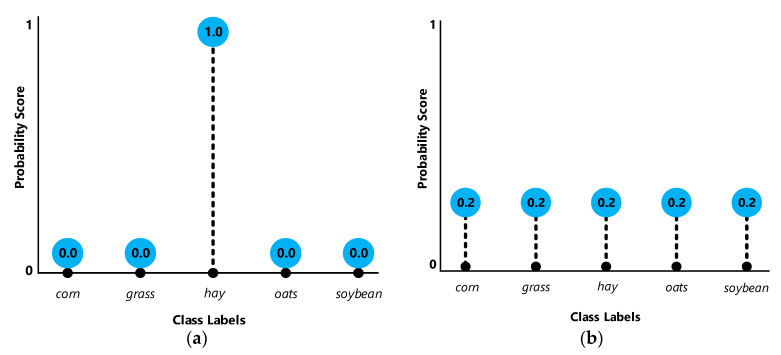
The best-case scenarios for every pixel representing low uncertainty (**a**) versus the worst-case scenario denoting high uncertainty (**b**). The other instances would be intermediate states of these two.

**Figure 3 entropy-21-00078-f003:**
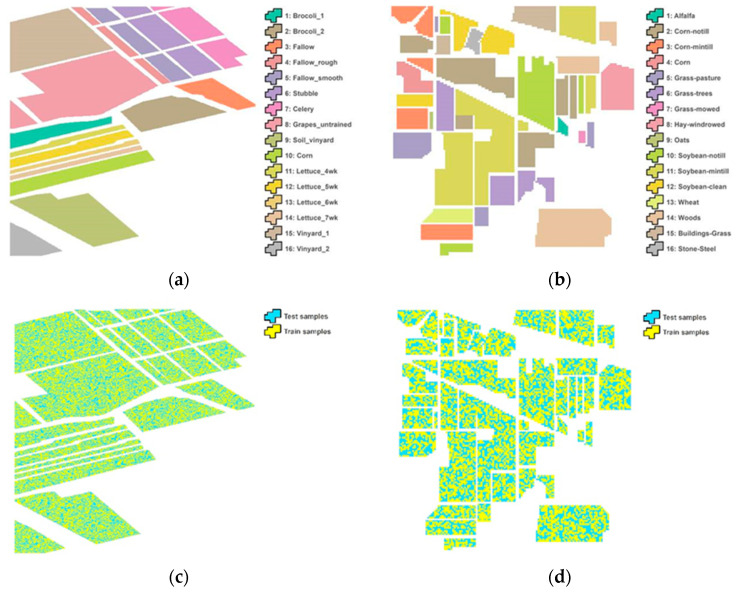
Ground truth data of two datasets including the Salinas (**a**) and the Indian Pines (**b**). The bottom images represent the location of the train and test data for the Salinas (**c**) and the Indian Pines (**d**).

**Figure 4 entropy-21-00078-f004:**
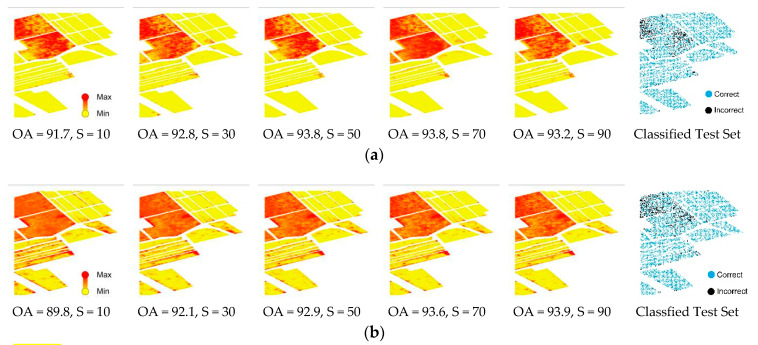
Results of uncertainty assessment for DNN (**a**) and RF (**b**) using different portions of training sample (S, in %) and mode of correct/incorrect classified test data for the Salinas dataset. The estimated overall accuracy (OA, in %) of the whole classification scheme is also demonstrated for each training sample.

**Figure 5 entropy-21-00078-f005:**
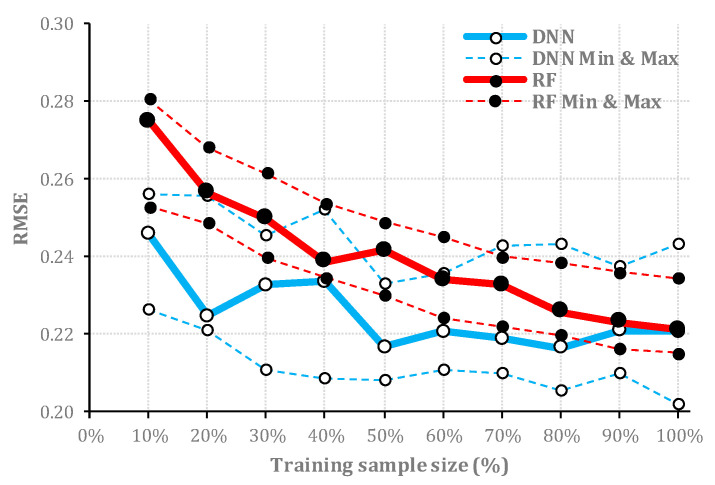
The estimated RMSE values of uncertainty assessment for test datasets (*y*-axis) where the algorithm is trained with different portions of the training sample (*x*-axis) of Salinas dataset. Dashed lines represent the minimum and maximum RMSE values for each sample size achieved in five consecutive simulation runs.

**Figure 6 entropy-21-00078-f006:**
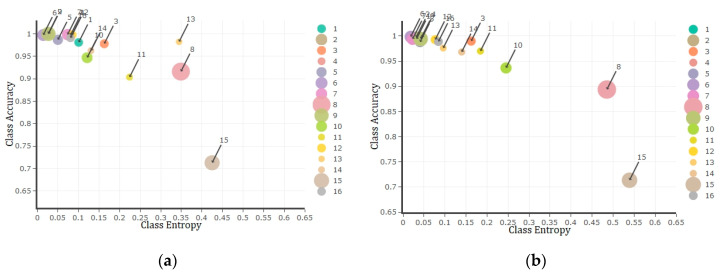
Class entropy/uncertainty (*x*-axis) versus class accuracy (*y*-axis) plots of Salinas dataset using DNN (**a**, left) and RF (**b**, right) algorithms observed by applying 50% of training data. The bubble sizes represent the frequency of land use class labels while bigger bubbles indicate the higher frequency and vice versa.

**Figure 7 entropy-21-00078-f007:**
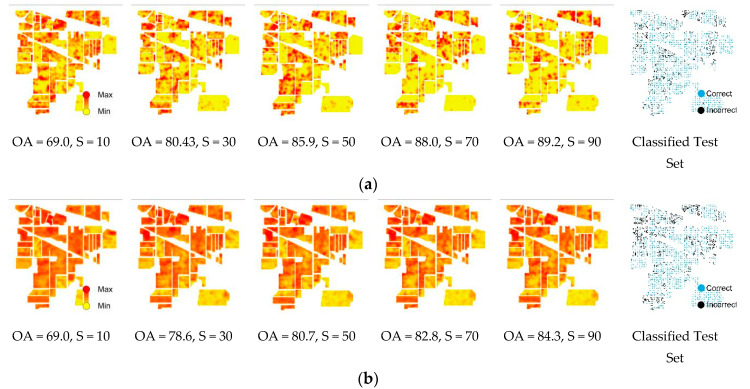
Results of uncertainty assessment for DNN (**a**) and RF (**b**) using different portions of training sample (S, in %) and mode of correct/incorrect classified test data for the Indian Pines dataset. The estimated overall accuracy (OA, in %) of the whole classification scheme is also demonstrated for each training sample.

**Figure 8 entropy-21-00078-f008:**
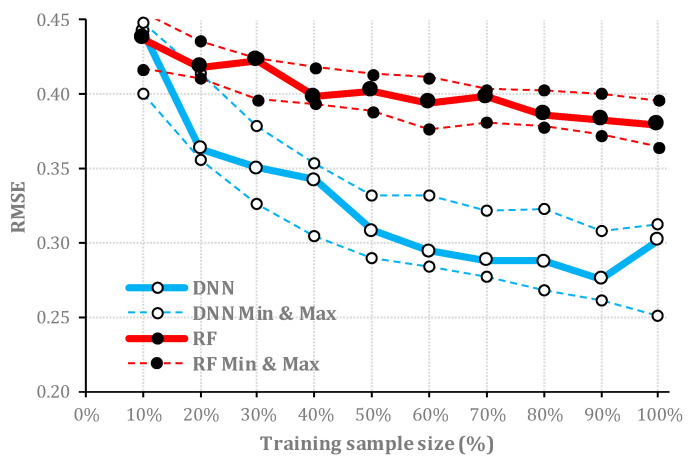
The estimated RMSE values of uncertainty assessment for test datasets (*y*-axis) where the algorithm is trained with different portions of training sample (*x*-axis) of Indian Pines dataset. Dashed lines represent the minimum and maximum RMSE values for each sample size achieved in five consecutive simulation runs.

**Figure 9 entropy-21-00078-f009:**
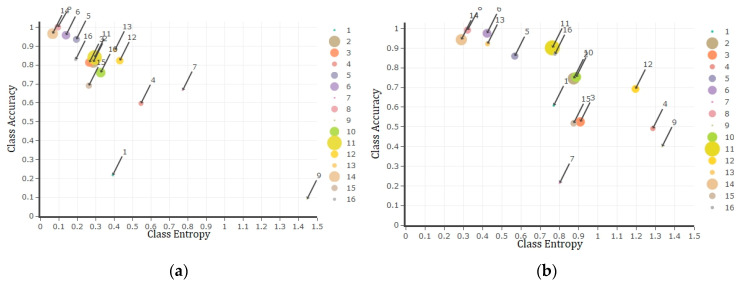
Class entropy/uncertainty (*x*-axis) versus class accuracy (*y*-axis) plots of Indian Pines dataset using DNN (**a**, left) and RF (**b**, right) algorithms observed by applying 50% of training sample size. The bubble sizes represent the frequency of land use class labels while bigger bubbles indicate the higher frequency and vice versa.

**Table 1 entropy-21-00078-t001:** The optimised hyper-parameters of DNN and RF using 5-fold cross-validation data for uncertainty assessment.

Algorithm	Hyper-Parameter	Description	Salinas	Indian Pines
DNN	hidden	Hidden layer sizes	(100, 100)	(200, 200)
DNN	epoch	How many times the dataset should be iterated (streamed)	300	300
DNN	activation	Activation function for non-linear transformation.	“Maxout”	“Maxout”
DNN	stopping metric	A metric that is used as a stopping criterion	“RMSE”	“RMSE”
DNN	l1	Only allows strong values to survives	0.0001	0.0001
DNN	l2	Prevents any single weight from getting too big	0.001	0.001
DNN	epsilon	Prevents getting stuck in local optima	1 × e^−10^	1 × e^−10^
RF	ntree	Number of trees to grow	100	100
RF	mtry	Number of variables available for splitting at each tree node	14	15

* For deep learning, this optimisation is done using “Grid Search” by h20.grid() function, and for random forest it has been done manually for the number of trees while tunerf() function is used to optimise mtry.

**Table 2 entropy-21-00078-t002:** The intuition behind the proposed RMSE.

Best-Case Scenarios	*e*	*o*	RMSE	Worst-Case Scenarios	*e*	*o*	RMSE
Positive	0	0	0	Positive	0	1	1
Negative	1	1	0	Negative	1	0	1

* All other instances fall within intermediate states.

**Table 3 entropy-21-00078-t003:** The major attributes of the hyperspectral datasets.

Dataset	Sensor	Total Bands	Excluded Bands	Number of Classes	Dimension	Resolution
**Salinas**	AVIRIS	224	20	16	512 × 217	20 metre
**Indian Pines**	AVIRIS	224	24	16	145 × 145	20 metre

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
