# Peer review of "Uncertainty Assessment of Hyperspectral Image Classification: Deep Learning vs. Random Forest"

_entropy, 2019, doi:10.3390/e21010078_

Round 1

Reviewer 1 Report

Good:  1. In-depth analysis.  Bad: 1. The entire analysis is tangential to the theme of the journal 2. The article could be significantly reduced in size by eliminating redundancy in discussion 3. It reads more like a dissertation and less like a paper.

Author Response

Dear Reviewer,

First of all, we would like to appreciate your time and efforts that you have dedicated to review this paper and suggesting minor but important changes for its improvement. We did our best to address all the comments provided, in order to bring the level of improvement aimed by the comments/suggestions.

Please find detailed responses to the comments below.

Warm regards,

Authors

CM #1. Good:  1. In-depth analysis.

Thanks a lot for this encouraging comment. This manuscript is part of PhD thesis and thus we did our best to do an in-depth analysis of the topic.

---------------------------------------------------------------------------------------------------------------------------

CM #2. Bad: 1. The entire analysis is tangential to the theme of the journal 2. The article could be significantly reduced in size by eliminating redundancy in discussion 3. It reads more like a dissertation and less like a paper.

To properly respond to the criticisms mentioned in this comment we have provided a point by point response:

1.      We have submitted this paper to Entropy journal since the Shannon entropy estimation is the back-bone of uncertainty analysis in this paper. According to the stated aim and scope of the journal “development and/or application of entropy or information-theoretic concepts in a wide variety of applications” are within the area of interest and encouraged for publication.  In addition, Shannon entropy applications are among the mentioned “Common Subject or Application Areas”, please see: https://www.mdpi.com/journal/entropy/about

Finally, we need to mention that this paper is submitted to a special issue of the journal entitled as “Entropy in Image Analysis” that is the main theme of our paper 9 (please see: https://www.mdpi.com/journal/entropy/special_issues/entropy_image_analysis ).

2.      We appreciate that you realised and mentioned that this paper is an in-depth analysis of the topic and aimed at contributing into the relevant literature. There is no doubt that a shorter paper is easier to read; however, more write-ups in this paper is aimed at clear communication of research funding. In addition, as it is announced by entropy journal at aim and scope section “There is no restriction on the length of the papers. If there are computation and the experiment, the details must be provided so that the results can be reproduced” (please see: https://www.mdpi.com/journal/entropy/about). Nonetheless, to improve the quality of the paper following your comment we have shortened the discussion section as much as possible. Those removed parts can be seen in a track change mode.

3.      Thank you for your comment. We have shortened the discussion as much as possible. Please see below:

“4. Discussion

4.1. Comparing the quality of uncertainty assessment based on RMSE

With reference to the fact that both DNN and RF algorithms may achieve an overall accuracy above 70%, even for the minimum portion of training sample size (i.e. 10%), it was expected one algorithm may perform a better uncertainty assessment if it successfully limits the high uncertainty areas to the spatial vicinity of incorrectly classified pixels while highlighting the remaining areas as low uncertainty. This is regardless of achieved overall accuracy. Although the RMSE values derived from five consecutive runs of each algorithm indicate that results of uncertainty assessment using RF is more consistent compared with DNN. Nonetheless, comparing the results of uncertainty assessment, for both utilised datasets and every corresponding sample sizes demonstrates that areas of high uncertainty values were less abundant within the results of DNN algorithm compared with RF (Figure 4 and Figure 7). This may be due to the fact that DNN is optimized to reduce the difference between the predicted distribution and the true data generating distribution by minimizing the cross-entropy of the two probability distributions [65,66]. Therefore, the uncertainty assessment derived from DNN algorithm was a superior to RF combined with a better overall accuracy for these two datasets. However, more studies using different datasets are still required for generalizing the results.

4.2. Quality of uncertainty assessment for different sample sizes

For both algorithms and both datasets, larger training samples were found to be more beneficial for uncertainty assessment. The RMSE of uncertainty estimates, which was applied as a goodness of fit to assess the quality of uncertainty maps, decreased from the initial (10%) to final (100%) training sample sizes (Figure 5 and 8). However, this improvement was more obvious for DNN compared with RF. This may be due to a different formulation of RF and DNN algorithms, which is affecting the performance of the two algorithms for uncertainty assessment. Usually, the training sample size has a crucial role in classification accuracy [67]; thus it will also affect the uncertainty assessment process. The increased training sample size will typically increase the performance of an algorithm from random sampling [68,69], but not all algorithms will be improved at the same level with a larger sample size. Although RF can also benefit from a larger training sample by extracting more binary rules [70], DNN may achieve a better performance. For DNN, the ratio of uncertainty assessment improvement followed by larger training sample size and more accurate classification depends on the abundance of contextual information per-pixel in the target dataset [71]. As many extensive experimental results confirm the excellent performance of the deep learning-based algorithms matched with rich spectral and contextual information [72], our study suggests this is also beneficial to increase training sample to achieve a better uncertainty assessment result.

4.3. Uncertainty vs accuracy

The existing uncertainties at different stages of the classification procedure influence classification accuracy [67,73]. Therefore, understanding the relationships between the classification uncertainty and accuracy is the key successful contribution of an estimate of accuracy for image classification. Although low uncertainty classification instance is accompanied with high accuracy, some exceptions may apply to the high uncertainty, which usually belongs to low accuracy estimates. Thus, incorrect predicted class labels are usually located inside high uncertainty areas with very few exceptions within low uncertainty regions while correct classified pixel overlay the low uncertainty areas (Figure 4 and Figure 7). In this research, for both applied datasets, the existing correspondence between uncertainty and accuracy was better identified using DNN algorithm. Having said that, in our study, DNN is demonstrated more potential in uncertainty assessment for hyperspectral image classification. Following accurate classifications combined with minimising high uncertainty areas, DNN not only offers a lower rate of RMSE but also offers a higher contrast between low and high uncertainty areas.

At a wider scale, considering mean class uncertainty against the class accuracy of test data, it was revealed that usually a lower uncertainty value of a class is followed by a higher accuracy (Figure 6 and Figure 9). In other words, as low uncertainty indicates the probabilities of potential class labels for a pixel are not equal (i.e. unimodal distribution). This simply specifies that based on the available distribution of potential labels and their probability values (Figure 2), defined by either deep learning neurons or tree votes, usually one of the potential class labels (i.e. 16 labels for each applied datasets) has a significant preference to be selected as the estimated label. Accordingly, the concentration of low uncertainty values corresponding to every pixels of a desired class label is anticipated by an acceptable accuracy of classification. In terms of higher values of mean uncertainty for a class, the class accuracy will be reduced due to abundance of high uncertainty estimates within that class. “

---------------------------------------------------------------------------------------------------------------------------

Thank you and all the very best wishes,

Authors

Reviewer 2 Report

The manuscript entitled "Uncertainty assessment of hyperspectral image classification: deep learning vs random forest" compares the performance of deep neural network with that of random forest concerning land-use imagery. It estimates the accuracy using the Shannon entropy and the quality of uncertainty assessment based on RMSE. Besides, it assesses the quality of uncertainty for different sample sizes, and compares uncertainty versus accuracy. The text is well written and well organized, the scientific question is sound, the methodology is robust, the results are well discussed, and the subject is of interest for the Journal readership. Therefore, we understand that the paper has high standard and should be accepted for publication in this Journal with minor review.

Please see below some suggestions, aiming at improving the manuscript.

Methodology. Please state where the Salinas and the Indian Pines are located and the timespan of the images. It would also be meaningful to present some basic description of the vegetative cover of the areas, as well as eventual remarks on the ground-truthing campaign (in which season and/or under which circumstances it was performed). This will help us to better understand the constraints of the work (LL 193 - 202). Besides, it should help explaining some conclusive statements, such as "DNN algorithm was more obvious when applying Indian Pines dataset" (why?).

Figure 5. Please add the meaning of the red and blue lines in the legend.

"RMSE values derived from five consecutive simulation run of RF are more consistent" (LL 262-263). This result is not straightforward from Figure 8. Please explain it more thoroughly.

Author Response

Reviewer 2:
Dear Reviewer,
Thank you for the time that you have spent on this paper.
We do appreciate your comments and your encouraging conclusion about the quality of the paper.
Regards,
Authors
Comments to the Author
CM #1. The manuscript entitled "Uncertainty assessment of hyperspectral image classification: deep learning vs random forest" compares the performance of deep neural network with that of random forest concerning land-use imagery. It estimates the accuracy using the Shannon entropy and the quality of uncertainty assessment based on RMSE. Besides, it assesses the quality of uncertainty for different sample sizes, and compares uncertainty versus accuracy. The text is well written and well organized, the scientific question is sound, the methodology is robust, the results are well discussed, and the subject is of interest for the Journal readership. Therefore, we understand that the paper has high standard and should be accepted for publication in this Journal with minor review.
Dear reviewer, this comment is appreciated and it is indeed motivating and encouraging.
---------------------------------------------------------------------------------------------------------------------------
CM #2. Methodology. Please state where the Salinas and the Indian Pines are located and the timespan of the images. It would also be meaningful to present some basic description of the vegetative cover of the areas, as well as eventual remarks on the ground-truthing campaign (in which season and/or under which circumstances it was performed). This will help us to better understand the constraints of the work (LL 193 - 202). Besides, it should help explaining some conclusive statements, such as "DNN algorithm was more obvious when applying Indian Pines dataset" (why?).
Dear reviewer, we have considered this comment and applied relevant changes in the manuscript as below: “The Salinas image consists of 224 bands and each band contains 512 × 217 pixels covering 16 classes composing different sub-classes of vegetables (nine sub-classes), bare soils (three sub-classes) and vineyard (four sub-classes). It was recorded by AVIRIS sensor over South of the city of Greenfield in the Salinas Valley, CA, USA on October 9, 1998. This dataset is characterised by a spatial resolution of 3.7 m, and the spectral information ranges from 0.4 to 2.5 μm. As shown in Error! Reference source not found., the ground truth is available for nearly two thirds of the entire scene. We used 204 bands, after removing bands of the water absorption features. The Indian Pines dataset is also an AVIRIS image collected over the Indian Pines test site location, Western Tippecanoe County, Indiana, USA on June 12, 1992. This dataset consists of 220 spectral bands in the same wavelength range as the Salinas dataset; however, four spectral bands are removed as they contain no data. This scene is a subset of a larger scene and it contains 145 × 145 pixels covering 16 ground truth classes (Error! Reference source not found.). The ground-truthing campaign consists of approximately 10,000 samples over which samples are distributed: 2.5 km by 2.5 km. The ground truth data were collected by walking through the fields in the image. Plant species, as well as some more characteristics, were recorded along with photos of sites in the field. In the present research experiment, 20 spectral bands were removed because of the water absorption phenomena and noise.”
Highlighted text in green is intended to emphasise applied changes.
---------------------------------------------------------------------------------------------------------------------------
CM #3. Figure 5. Please add the meaning of the red and blue lines in the legend.
This comment is considered in both Figures 5 and 8 as below:
Figure 1. The estimated RMSE values of uncertainty assessment for test datasets (y-axis) where the
algorithm is trained with different portions of the training sample (x-axis) of Salinas dataset. Dash lines
represent the minimum and maximum RMSE values for each sample sizes achieved in five consecutive
simulation run.
Figure 2. The estimated RMSE values of uncertainty assessment for test datasets (y-axis) where the
algorithm is trained with different portions of training sample (x-axis) of Indian Pines dataset. Dash
lines represent the minimum and maximum RMSE values for each sample sizes achieved in five
consecutive simulation run.
---------------------------------------------------------------------------------------------------------------------------
CM #4. "RMSE values derived from five consecutive simulation run of RF are more consistent" (LL
262-263). This result is not straightforward from Figure 8. Please explain it more thoroughly.
This comment is considered and we have added some explanations as below:
0.20
0.22
0.24
0.26
0.28
0.30
0% 10% 20% 30% 40% 50% 60% 70% 80% 90% 100%
RMSE
Training sample size (%)
DNN
DNN Min & Max
RF
RF Min & Max
0.20
0.25
0.30
0.35
0.40
0.45
0% 10% 20% 30% 40% 50% 60% 70% 80% 90% 100%
RMSE
Training sample size (%)
DNN
DNN Min & Max
RF
RF Min & Max
Page 9, Line 266-271:
“However, the same as Salinas dataset, RMSE values derived from five consecutive simulation run of RF are more consistent. This can be easily observed by comparing the difference between the minimum and maximum RMSE values for each sample size that is observable in Figure 8. Obviously, DNN is coupled with more variation between minimum and maximum RMSE values for almost all different sample sizes.”
---------------------------------------------------------------------------------------------------------------------------
Thanks, and all the very best wishes,
Authors
